# Impact of COVID-19 pandemic on breast cancer screening in a large midwestern United States academic medical center

**Kimberly J. Johnson**[1,2]*, **Caitlin P. O'Connell**[1], **R. J. Waken**[3,4], **Justin M. Barnes**[5]

**1** Brown School, Washington University in St. Louis, St. Louis, Missouri, United States of America, **2** Siteman Cancer Center, Washington University in St Louis, St. Louis, Missouri, United States of America, **3** Division of Biostatistics, Washington University School of Medicine, St. Louis, Missouri, United States of America, **4** Center for Advancing Health Services, Policy & Economics Research, Washington University School of Medicine, St. Louis, Missouri, United States of America, **5** Department of Radiation Oncology, Washington University School of Medicine, St. Louis, Missouri, United States of America

* kijohnson@wustl.edu

**Data Availability Statement:** De-identified or anonymized data cannot be shared publicly because data contain potentially identifying or sensitive patient information. Data are available

## Abstract

### Background

Access to breast screening mammogram services decreased during the COVID-19 pandemic. Our objectives were to estimate: 1) the COVID-19 affected period, 2) the proportion of pandemic-associated missed or delayed screening encounters, and 3) pandemic-associated patient attrition in screening encounters overall and by sociodemographic subgroup.

### Methods

We included screening mammogram encounter EPIC data from 1-1-2019 to 12-31-2022 for females ≥40 years old. We used Bayesian State Space models to describe weekly screening mammogram counts, modeling an interruption that phased in and out between 3-1-2020 and 9-1-2020. We used the posterior predictive distribution to model differences between a predicted, uninterrupted process and the observed screening mammogram counts. We estimated associations between race/ethnicity and age group and return screening mammogram encounters during the pandemic among those with 2019 encounters using logistic regression.

### Results

Our analysis modeling weekly screening mammogram counts included 231,385 encounters (n = 127,621 women). Model-estimated screening mammograms dropped by >98% between 03-15-2020 and 05-24-2020 followed by a return to pre-pandemic levels or higher with similar results by race/ethnicity and age group. Among 79,257 women, non-Hispanic (NH) Asians, NH Blacks, and Hispanics had significantly (p < .05) lower odds of screening encounter returns during 2020–2022 vs. NH Whites with odds ratios (ORs) from 0.70 to 0.91. Among 79,983 women, those 60–69 had significantly higher odds of any return screening encounter during 2020–2022 (OR = 1.28), while those ≥80 and 40–49 had

from the Institute for Clinical and Translational Sciences Informatics Core Services (https://icts.wustl.edu/research-services/informatics/) for researchers who meet the criteria for access to confidential data. For more information on sharing research data see "Sharing Research Data" in the Washington University in St. Louis Human Research Protections Research Guide https://online.fliphtml5.com/ikcz/ifub/#p=154. See https://hrpo.wustl.edu/about-us/contact-us/ for Institutional Review Board contact information for questions.

**Funding:** This work was supported by a just-in-time award through NIH CTSA Grant #UL1 TR002345 and through the Neidorff Family and Centene Corporation COVID & Health Disparity Response Fund. The funders had no role in study design, data collection and analysis, decision to publish, or preparation of the manuscript.

**Competing interests:** The authors have declared that no competing interests exist.

significantly lower odds (ORs 0.77, 0.45) than those 50–59 years old. A sensitivity analysis suggested a possible pre-existing pattern.

## Conclusions

These data suggest a short-term pandemic effect on screening mammograms of ~2 months with no evidence of disparities. However, we observed racial/ethnic disparities in screening mammogram returns during the pandemic that may be at least partially pre-existing. These results may inform future pandemic planning and continued efforts to eliminate mammogram screening disparities.

## Introduction

Cancer is the leading cause of death in the United States with >500,000 deaths per year [1]. Morbidity and mortality from some cancer types, including breast cancer, are preventable through screening and early-stage treatment [2]. It is well understood that diagnosis delays can result in an increased risk of death and long-term morbidity due to the difficulty of curing cancer at advanced stages and the need for more aggressive treatment with data suggesting pandemic-associated shifts in cancer stage [3].

The COVID-19 pandemic resulted in reductions in non-essential services including reports of surgery delays for cervical cancer [4], temporary stoppages of cancer screening [5], and cancellation of visits to providers that diagnose cancer (e.g. women's clinics) [6]. Reductions in screening could lead to non-guideline concordant screening aimed at preventing later-stage breast cancer diagnoses. Two of the major guideline bodies that make breast cancer screening recommendations in the United States are the American Cancer Society (ACS) and the United States Preventive Services Task Force (USPSTF). The ACS recommends that women ages 40–44 years have the option to start annual screening with annual screening recommended for women ages 45–54 years and biennial screening at 55 years [7]. The USPSTF currently recommends (with draft changes under consideration [8]) optional screening for women ages 40–49 years with biennial screening recommended for women ages 50–74 years [9].

Mammograms have been reported to have decreased early in the pandemic [10] with evidence for larger drops in screening in women who are Black, Hispanic, and American Indian/Alaskan Native and who are from lower socioeconomic groups [11–14]. In addition, patients may have been slow to return to preventative cancer care due to fear of being infected at health facilities, especially older patients who may be more vulnerable to morbidity and mortality from COVID-19.

To add to the literature, we used using a Bayesian State Space modeling approach [15] to analyze temporal trends in mammography screening from a large academic medical center in the midwestern United States from 2019 through 2021. This method overcomes some of the limitations of prior studies comparing pre-pandemic to pandemic mammogram screening counts. We also examined changes in return encounters for screening at different intervals during the pandemic. Specifically, our objectives were to: 1) quantify the COVID-19 affected period overall and by race/ethnicity and age group, 2) estimate the proportion of pandemic-associated missed or delayed visits overall and by race/ethnicity and age group, and 3) estimate the odds of returning for screening mammograms during the pandemic to understand pandemic-associated attrition among the cohort screened in the year before the pandemic by race/ethnicity and age group.

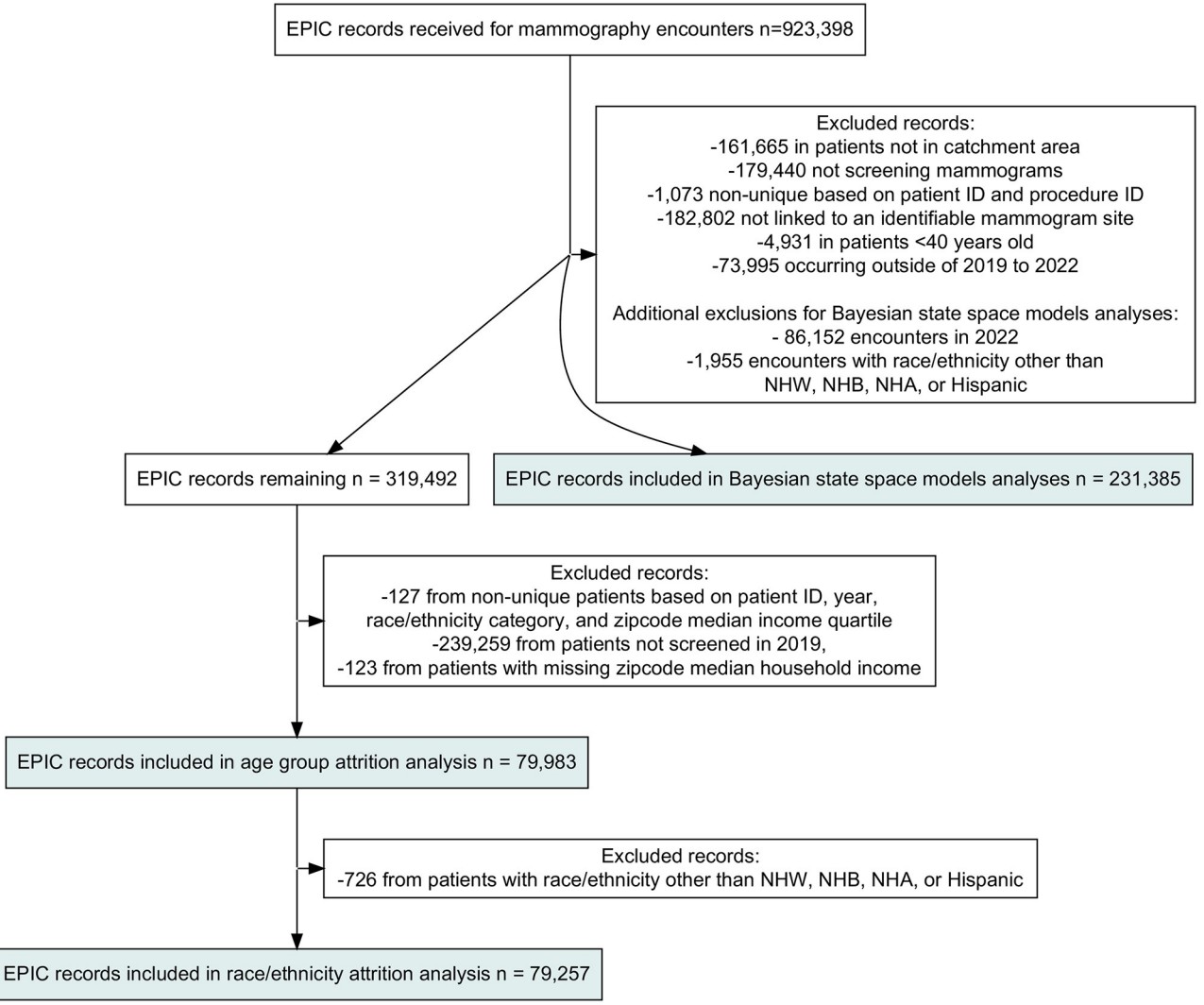

**Fig 1. Flow chart of exclusions.**

To conduct these analyses, we obtained data on over 300,000 encounters for screening mammogram exams during a four-year period. This work builds on earlier work with a longer period of observation using an analytic method not previously used and contributes to our understanding of pandemic-associated breast cancer screening trends and disparities in women.

## Methods

### Study population

The study population includes women with screening mammogram encounters conducted within the Barnes Jewish Christian (BJC) Healthcare System between 1-1-2019 and 12-31-2022 among women who were ≥40 years old at the time of their encounter. The data containing identifying information was extracted from the BJC EPIC database on 3-24-2023 for research purposes. For sensitivity analyses, the same criteria were applied except we used screening encounter data from 6-2-2018 (the earliest date available) to 12-31-2019. See Fig 1

and Supporting information for detailed methods on exclusions leading to our analytic data-sets. The Washington University in St. Louis Institutional Review Board approved this study.

## Variables

Age at the screening mammogram encounter was calculated as the difference between the birth date and the screening mammogram encounter date. We created age group categories of 40–49, 50–59, 60–69, 70–79, and $\geq$ 80 years old. Race and ethnicity were captured in EPIC as Hispanic, White, Black or African American, Asian, American Indian or Alaska Native, and Other Pacific Islander with the ability for multiple races to be selected. We categorized each race with regard to ethnicity as non-Hispanic (NH) and Hispanic. We created an indicator variable for the early pandemic that was set to one if the mammogram screening encounter date was between 3-1-2020 and 8-31-2020, otherwise, it was set to zero. We also created an indicator variable for holidays to account for the effect that holidays have on encounter counts per week. The included holidays were: New Year's Day, Memorial Day, Independence Day, Labor Day, Thanksgiving Day, and Christmas Day if they fell during the week. For the attrition analysis, we defined four dependent variables to categorize mammogram screening encounter patterns among women with consideration for screening guidelines that vary by age group. The first variable, 'annual screening encounter return in 2020', is a binary variable defined as having vs. not having a screening encounter in the data in 2020. The second variable, 'biennial screening encounter return in 2021' is a binary variable defined as having vs. not having a screening encounter in 2021 but not 2020 in the data. The third variable, 'triennial screening encounter return in 2022 ' is a binary variable defined as having vs. not having a screening encounter in 2022 but not in 2020 or 2021 in the data. The fourth variable, 'Any screening encounter return in 2020–2022' is a binary variable defined as having vs. not having any screening encounter in 2020–2022 in the data. Of note, we only used year and not date to classify individuals according to annual, biennial, and triennial screening. For example, women who were screened in 2019 were classified as having an annual screen in 2020 if they had a screening encounter recorded at any point in 2020 which could be up to almost two years later if they were screened for example on 1-1-2019 and returned for a screen on 12-31-2020. For the attrition sensitivity analysis (described further below), we created a binary variable, 'annual screening encounter return in 2019' to categorize screening patterns among women screened from 6-2-2018 to 12-31-2018 defined as having vs. not having a screening encounter in the data in 2019 among women screened during the 2018 period. This analysis was conducted to assess patterns existing before the pandemic. "Median Household Income" data for zip codes in 2021 inflation-adjusted dollars was extracted from Social Explorer [16]. A total of 195 of the 247 zip codes in the BJC catchment area had median household income data available in the Social Explorer file. We created median income zip code quartile categories for zip codes in the catchment area. Individuals were assigned a median household income category based on the zip code of their residence during their 2019 encounter for the main analyses and their 2018 encounter for the sensitivity analysis. See Supporting information for zipcodes and social explorer data used in this study.

## Statistical analysis

All analyses were conducted in R version 4.2.3. or 4.3.0 and Stan version 2.21.0. All estimates were considered statistically significant if p < .05.

**Bayesian State Space modeling analysis.** To account for trends and non-independence over time, a Bayesian state-space model [15] was fit to describe screening mammogram counts in the groups of interest while incorporating spectral trends (cyclical patterns) and holiday

effects. Women with screening mammogram encounters in 2019, 2020, and/or 2021 were included. Details on the prior parameterization and the state portion, or the portion of the model governing the evolution of the parameters over time, are provided in the **Supporting information**. We fit a log-normal Poisson regression model as:

$$\log(E(y_{it})) = \mu_{it} + H_t \times \beta_H + S_t \times \beta_S,$$

where t is the time index, i is the index for a group observation at time t, $y_{it}$ is the expected screening mammogram count for a group at time t, $\mu_{it}$ is the mean process effect attributed to group $i$ at time $t$, $H_t$ and $\beta_H$ are the holiday dummy indicator and effects respectively, and $S_t$ and $\beta_S$ are the seasonality covariate and terms respectively. At each time $t$ within each group $i$, $\mu_t$ is calculated as:

$$\mu_t = \tau_t X_t^T \beta_t + (1 - \tau_t) X_t^T \omega_t,$$

where $X_t^T \beta_t$ is the time-varying mean process with no acute pandemic effect, $X_t^T \omega_t$ is the time-varying mean process that represents the depressed mean process from 3-1-2020 to 8-31-2020, and $\tau_t \in (0, 1)$ is the averaging parameter that weights the two mean processes when making predictions. Although the system paused regular screening mammograms officially on 3-23-2020 [17] with resumption beginning in early May, healthcare utilization avoidance by patient choice was prevalent during the pandemic [18], and the data-driven modeling procedure above allows the mean number of screening encounters to drop off and return based on encounter data rather than assuming an immediate pause and return based on when administrative guidance was distributed. All elements of the mean function aside from the seasonal and holiday effects were allowed to vary within group as all baseline rates, avoidance, timing, and the overall drop in utilization may have differed by group.

*Posterior inferences.* We used the posterior predictive distribution to predict counts of screening mammogram encounters using the above state-space model with interval estimates derived from Markov chain Monte Carlo (MCMC) samples of the parameters of interest. Ninety-five percent credible intervals are derived from the 2.5th and 97.5th percentiles of MCMC draws [19]. All p-values comparing group rate changes in the $\omega$ were computed using a posterior-based Wald-type test statistic [20].

*Pandemic effects on encounters.* To predict the number of dropped visits, we generated a $(1 - \pi_t)$ weighted sums as well as weighted quantiles of the difference

$$\hat{y}'_{it} = \hat{y}^o_{it} - \hat{y}_{it},$$

where $\hat{y}_{it}$ is the predicted quantity from the full model, and $\hat{y}^o_{it}$ are the predictions generated solely by the portion of the model under no acute pandemic effect. The use of the state space time-varying regression model allows us to create a Brownian bridge [21] from the pre-pandemic trend to the post-acute pandemic trend to create the predicted, unobserved $\hat{y}^o_{it}$. By comparing $\hat{y}^o_{it}$ to $\hat{y}_{it}$ to predict drops in utilization, we account for prediction errors in the pandemic onset timing from $\pi_{it}$ as well as the mean total drop effect modeled by $\omega_{it}$, and compare this estimated quantity to the forecasted and backcasted prediction in a time-varying model with baseline differences in groups, holiday effects, and seasonality terms. To prevent potential label switching issues and reduce computational burden, we allow the model to select any date in the window on or after 3-01-2020 before 4-15-2020 to start the pandemic acute drop, and any point between 4-15-2020 and 8-1-2020 to end the pandemic period.

For details on time-varying parameter evolution, prior parameterization, and specifics regarding how the parameter governing averaging between the acute pandemic and no acute pandemic mean function is modeled, see Supporting information.

**Attrition analysis.**   We used binary logistic regression to estimate odds ratios (ORs) and 95% confidence intervals (CIs) to model the odds of a screening encounter as defined by the dependent variable categories described above in association with race/ethnicity and age group. In models where race/ethnicity was the exposure of interest, we adjusted for median household income quartile category and the individual's age category in 2019. In models where age group was the exposure of interest, we only adjusted for median household income quartile category. We included only women who had 2019 mammogram screening encounters in our main analyses. The age group in 2019 was used to assign the age group for the analysis where age group was the exposure of interest. We also conducted a sensitivity analysis that determined the odds of a return annual mammogram screening encounter in 2019 among those screened in 2018 (from 6-2-2018 the earliest date that data was available through 12-31-2018) in association with race/ethnicity and age group. The age group in 2018 was used to assign the age group for these analyses.

## Results

### Characteristics of the study population

Out of 923,398 mammogram encounter records initially extracted from EPIC, subsequent exclusions resulted in our analytic datasets for our Bayesian State Space models, age group attrition, and race/ethnicity attrition analyses included 231,835 (127,621 women), 79,983 (79,983 women), and 79,257 (79,257 women) mammogram screening encounters (Fig 1). The average age at the screening encounter was similar across datasets and years at ~60 years. Encounters were most frequent among women who were 60–69 years old, NH White, married, residing in zip codes in the fourth quartile of median income, and who lived in Missouri, with similar patterns across datasets and years (Table 1).

### Model predicted periods where the pandemic affected mammogram encounters

There was a clear decrease in screening mammograms at the end of March through April 2020 (Fig 2). The model predicted period that the pandemic affected screening mammogram encounter counts was between 3-15-2020 and 5-24-2020 (Table 2). By race/ethnicity subgroup, variation in the predicted pandemic-affected period was observed starting later than the overall predicted period for NH Whites, NH Black or African-Americans, and Hispanics on 3-22-2020 and ending earlier for NH Black or African-American women on 5-17-2020 (Fig 3, Table 2). When comparing screened patients by age group, those <70 years old had a later predicted start date on 3-22-2020 than older women and the predicted end date was a week earlier than the overall end date (Fig 4, Table 2). Of note, the overall versus the subgroup end dates may differ because allowing the modeled probability of a pandemic pause in screenings to differ by subgroup reduces the overall variability in this probability estimate and because our mean and variance are dependent, this may change our best estimate for when the pause ended. Further, because the dates at which the pandemic pauses are considered to have ended are based on a confidence bound, reducing variability is also expected to change the times associated with a return to a normal process.

### Model predicted drops in screening mammogram encounters due to the pandemic

Compared to what would have been expected in the absence of a pandemic, the model predicted mean encounters was 11,543 equating to a 98.8% (95% CI 95.1% to 100%) drop in

**Table 1. Characteristics of women for encounters overall and by year for the time series and attrition analyses.**

| Variable | Bayesian State Space analyses | | | Attrition analyses | |
|---|---|---|---|---|---|
| | 2019 (N = 79,426) | 2020 (N = 68,449) | 2021 (N = 83,510) | 2019 age category cohort (N = 79,983) | 2019 cohort race/ethnicity (N = 79,257) |
| **Age (years)—mean (SD)** | 59.6 (11.0) | 59.9 (11.0) | 59.9 (11.2) | 59.6 (11.0) | 59.6 (11.0) |
| **Age group (years)** | | | | | |
| 40–49 | 16,950 (21.3%) | 14,490 (21.2%) | 18,197 (21.8%) | 17,163 (21.5%) | 16,918 (21.3%) |
| 50–59 | 22,217 (28.0%) | 18,370 (26.8%) | 21,706 (26.0%) | 22,412 (28.0%) | 22,175 (28.0%) |
| 60–69 | 24,276 (30.6%) | 21,156 (30.9%) | 25,472 (30.5%) | 24,384 (30.5%) | 24,220 (30.6%) |
| 70–79 | 13,250 (16.7%) | 12,052 (17.6%) | 15,160 (18.2%) | 13,286 (16.6%) | 13,217 (16.7%) |
| ≥80 | 2,733 (3.4%) | 2,381 (3.5%) | 2,975 (3.6%) | 2,738 (3.4%) | 2,727 (3.4%) |
| **Race/ethnicity** | | | | | |
| NH White | 58,171 (73.2%) | 49,750 (72.7%) | 60,110 (72.0%) | 58,091 (72.6%) | 58,091 (73.3%) |
| NH Black or African-American | 18,289 (23.0%) | 16,250 (23.7%) | 19,941 (23.9%) | 18,207 (22.8%) | 18,207 (23.0%) |
| NH Asian | 1,632 (2.1%) | 1,263 (1.8%) | 1,865 (2.2%) | 1,629 (2.0%) | 1,629 (2.1%) |
| NH American Indian or Alaska Native | 0 (0%) | 0 (0%) | 0 (0%) | 91 (0.1%) | 0 (0%) |
| NH Other Pacific Islander | 0 (0%) | 0 (0%) | 0 (0%) | 65 (0.1%) | 0 (0%) |
| NH Other | 0 (0%) | 0 (0%) | 0 (0%) | 120 (0.2%) | 0 (0%) |
| NH Two or more | 0 (0%) | 0 (0%) | 0 (0%) | 180 (0.2%) | 0 (0%) |
| Hispanic | 1,334 (1.7%) | 1,186 (1.7%) | 1,594 (1.9%) | 1,330 (1.7%) | 1,330 (1.7%) |
| Missing | 0 (0%) | 0 (0%) | 0 (0%) | 270 (0.3%) | 0 (0%) |
| **Marital status** | | | | | |
| Married | 47,092 (59.3%) | 40,985 (59.9%) | 49,623 (59.4%) | 47,464 (59.3%) | 47,033 (59.3%) |
| Single | 32,142 (40.5%) | 27,295 (39.9%) | 33,615 (40.3%) | 32,302 (40.4%) | 32,033 (40.4%) |
| Unknown/Missing | 192 (0.2%) | 169 (0.2%) | 272 (0.3%) | 217 (0.3%) | 191 (0.2%) |
| **Median income quartile for zip code of residence** | | | | | |
| Q1 | 16,156 (20.3%) | 13,848 (20.2%) | 17,483 (20.9%) | 16,269 (20.3%) | 16,142 (20.4%) |
| Q2 | 19,644 (24.7%) | 16,887 (24.7%) | 20,756 (24.9%) | 19,832 (24.8%) | 19,640 (24.8%) |
| Q3 | 15,120 (19.0%) | 13,163 (19.2%) | 15,816 (18.9%) | 15,240 (19.1%) | 15,110 (19.1%) |
| Q4 | 28,387 (35.7%) | 24,463 (35.7%) | 29,326 (35.1%) | 28,642 (35.8%) | 28,365 (35.8%) |
| Missing | 119 (0.1%) | 88 (0.1%) | 129 (0.2%) | 0 (0%) | 0 (0%) |
| **Patient's State of Residence** | | | | | |
| Illinois | 22,771 (28.7%) | 19,701 (28.8%) | 24,445 (29.3%) | 22,883 (28.6%) | 22,738 (28.7%) |
| Missouri | 56,655 (71.3%) | 48,748 (71.2%) | 59,065 (70.7%) | 57,100 (71.4%) | 56,519 (71.3%) |

Abbreviations: SD = standard deviation

expected encounters (Table 2). There was no significant variation in the percentage drop in expected visits by race/ethnicity (p = 1) or age group (p = .76) (Table 2, Fig 4).

## Attrition in screening mammogram encounter returns by race/ethnicity and age category

The odds of an annual screening mammogram return encounter in 2020 vs. none was the same at 1.00 (95% CI 0.97 to 1.04) in NH Black or African-Americans and lower in Hispanics (OR = 0.81; 95% CI 0.72 to 0.90) and NH Asians (OR = 0.68; 95% CI 0.62 to 0.76) vs. NH Whites. The odds of a biennial screening mammogram return encounter in 2021 vs. none in 2020 or 2021 was lower in NH Black or African-Americans (OR = 0.89; 95% CI 0.85 to 0.93), similar in Hispanics (OR = 1.04; 95% CI 0.92 to 1.18) and higher in NH Asians (OR = 1.18;

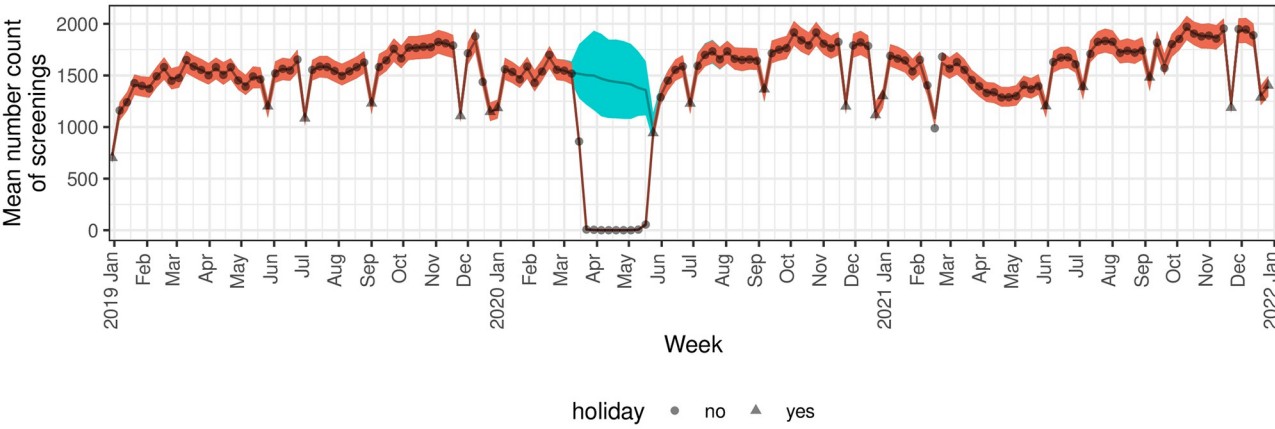

**Fig 2. Predicted vs. actual number of screening mammogram encounters by week and year.** Actual (gray dots and triangles) and predicted (brown line) number of mammograms per week by year model results. The orange area indicates 95% credible intervals and the blue is the prediction band for the situation where there was no pandemic in effect.

95% CI 1.05 to 1.32) compared to NH Whites. The odds of a triennial screening mammogram return encounter in 2022 vs. none in 2020–2022 was higher in all three racial/ethnic minority groups but was only significant in NH Black or African-Americans (OR = 1.10, 95% CI 1.02 to 1.19) compared to NH Whites. Finally, the odds of any screening encounter return vs. none during the pandemic years were significantly lower in all three racial/ethnic minority groups vs. NH Whites with ORs ranging from 0.70 to 0.91 (Table 3).

For the age group analysis, compared to 50-59-year-olds there were significantly higher odds of an annual screening mammogram return encounter in 2020 vs. none for those 60–79 years old ($OR_{60-69}$ = 1.21, 95% CI 1.17 to 1.26; $OR_{70-79}$ = 1.21, 95% CI 1.16 to 1.27) and significantly lower odds for those 40–49 and ≥80 years old ($OR_{40-49}$ = 0.83, 95% CI 0.80 to 0.86; $OR_{\geq 80}$ = 0.88, 95% CI 0.81 to 0.95). The odds of a biennial screening mammogram return

**Table 2. Screening encounter model estimates of the pandemic period and missed or delayed visits overall and by race and age group.**

| Group | Pandemic period first week | Pandemic period last week | Missed or delayed visits during the pandemic period N | Missed or delayed visits during the pandemic period % (95% CI[a]) | P-value |
|---|---|---|---|---|---|
| **Overall** | 3-15-2020 | 5-24-2020 | 11543.0 | 98.8 (95.1 to 100) | |
| **Race/ethnicity** | | | | | 1 |
| NH White | 3-22-2020 | 5-24-2020 | 8234.7 | 98.6 (90.8 to 100) | |
| NH Black or African-American | 3-22-2020 | 5-17-2020 | 2427.0 | 99.6 (96.4 to 100) | |
| Hispanic | 3-22-2020 | 5-24-2020 | 201.7 | 99.1 (93.8 to 100) | |
| NH Asian | 3-15-2020 | 5-24-2020 | 219.0 | 98.9 (95.1 to 100) | |
| **Age group (years)** | | | | | 0.76 |
| 40–49 | 3-22-2020 | 5-17-2020 | 2306.4 | 99.0 (94.7 to 100) | |
| 50–59 | 3-22-2020 | 5-17-2020 | 3235.8 | 99.0 (95.0 to 100) | |
| 60–69 | 3-22-2020 | 5-17-2020 | 3349.8 | 99.4 (96.2 to 100) | |
| 70–79 | 3-15-2020 | 5-17-2020 | 1994.8 | 99.1 (94.5 to 100) | |
| ≥80 | 3-15-2020 | 5-17-2020 | 394.6 | 99.1 (90.7 to 100) | |

[a]CI = Credible Interval

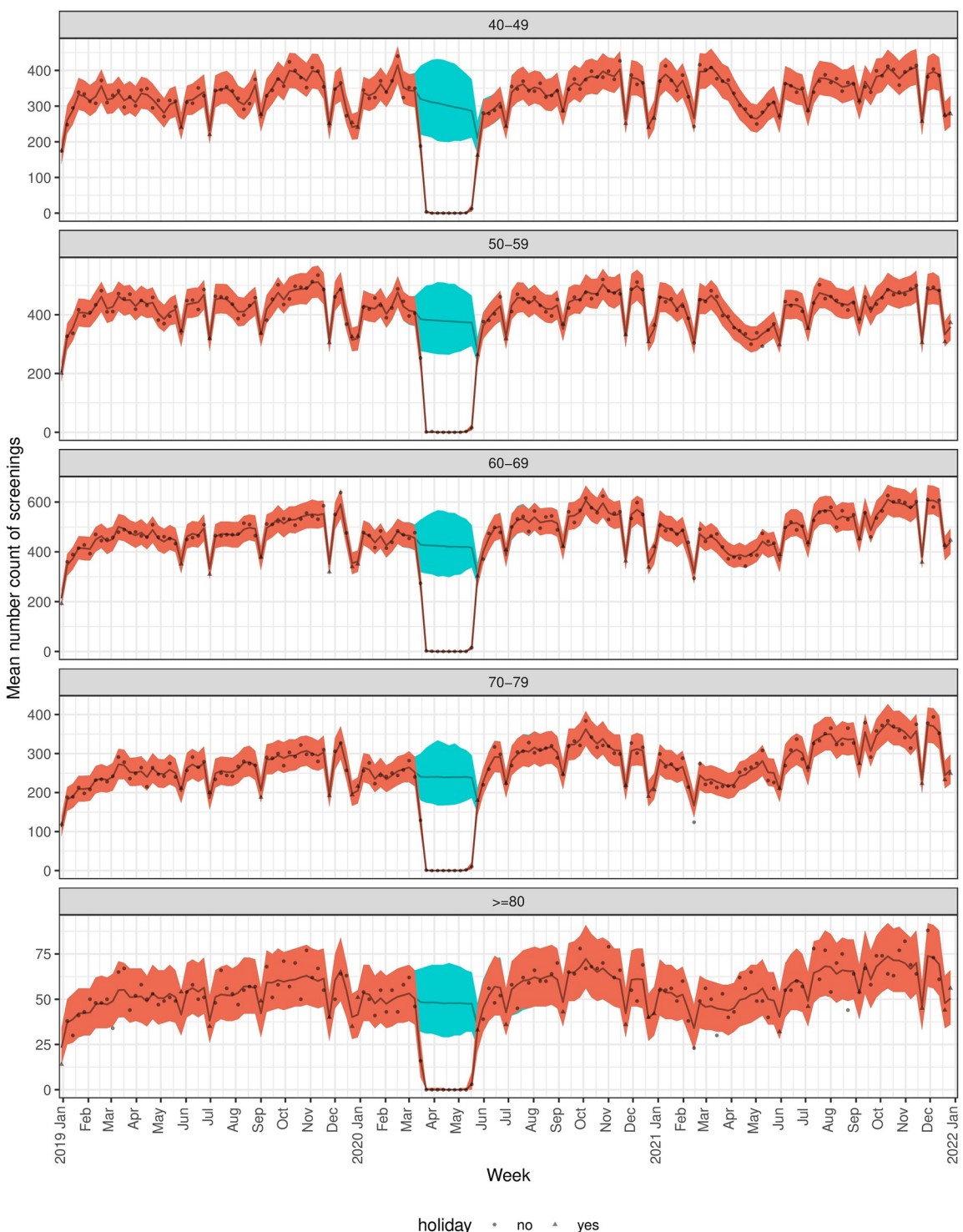

**Fig 3. Predicted vs, actual number of mammogram encounters by week stratified by race/ethnicity.** Actual (gray dots and triangles) and predicted (brown line) number of mammograms per week by year model results. The orange area indicates 95% credible intervals and the blue is the prediction band for the situation where there was no pandemic in effect.

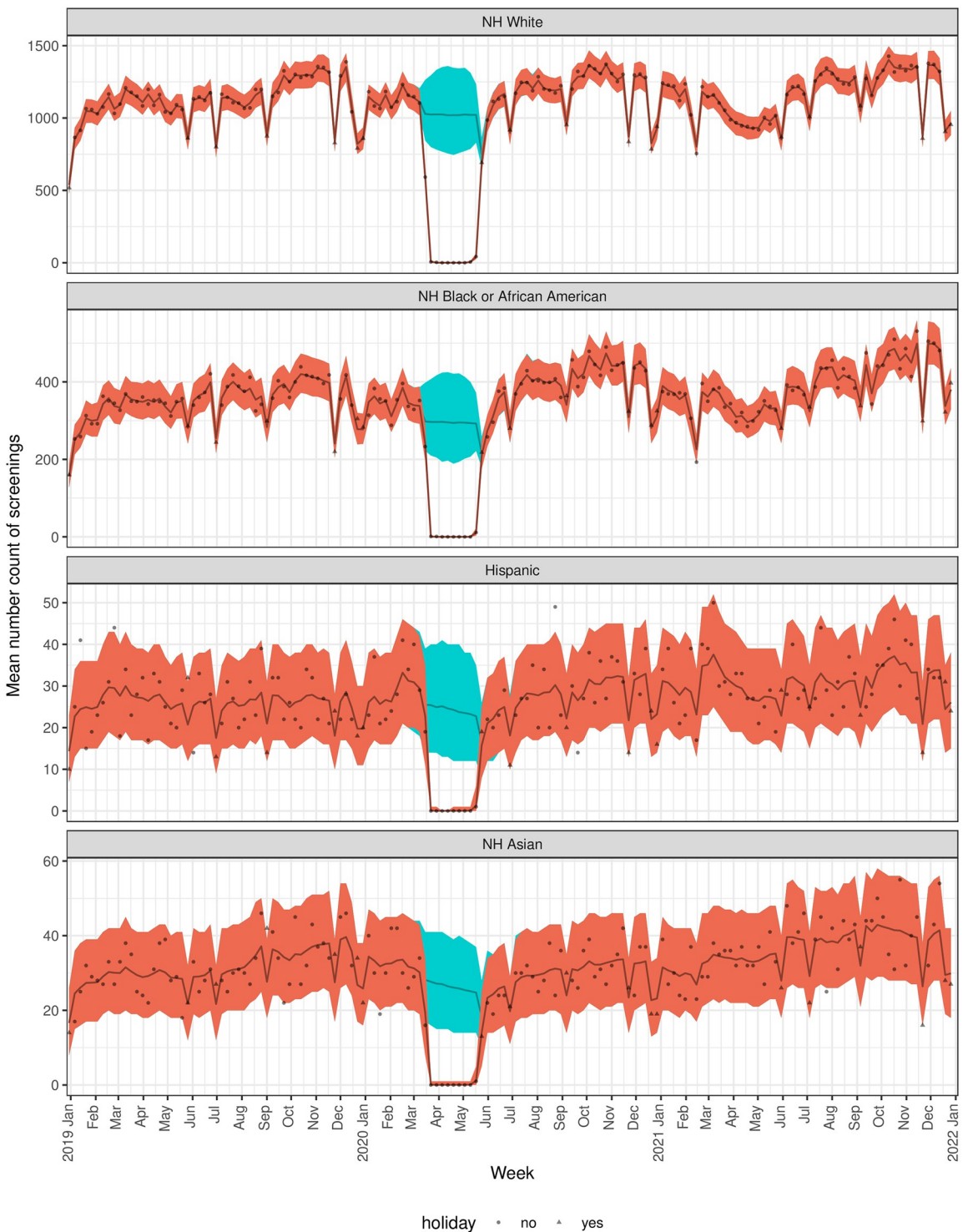

**Fig 4. Predicted vs. actual number of mammogram encounters by week stratified by age group.** Actual (gray dots and triangles) and predicted (brown line) number of mammograms per week by year model results. The orange area indicates 95% credible intervals and the blue is the prediction band for the situation where there was no pandemic in effect.

**Table 3. Race/ethnicity and age group associated attrition in mammogram screening among those with screening mammogram encounters in 2019 during the COVID-19 pandemic and among those with screening mammogram encounters in 2018 before the pandemic[a].**

| Variables | Annual screening encounter return in 2020[b,d] | Biennial screening encounter return in 2021[b,d] | Triennial screening encounter return in 2022[b,d] | Any screening encounter return in 2020–2022[b,d] | Annual screening encounter return in 2019[c,e] |
|---|---|---|---|---|---|
| Race/ethnicity | OR (95% CI) | OR (95% CI) | OR (95% CI) | OR (95% CI) | OR (95% CI) |
| NH White | 1.0 (reference) | 1.0 (reference) | 1.0 (reference) | 1.0 (reference) | 1.0 (reference) |
| NH Black or African-American | 1.00 (0.97 to 1.04) | 0.89 (0.85 to 0.93) | 1.10 (1.02 to 1.19) | 0.91 (0.86 to 0.95) | 0.95 (0.90 to 1.01) |
| Hispanic | 0.81 (0.72 to 0.90) | 1.04 (0.92 to 1.18) | 1.17 (0.94 to 1.44) | 0.79 (0.69 to 0.91) | 0.71 (0.60 to 0.84) |
| NH Asian | 0.68 (0.62 to 0.76) | 1.18 (1.05 to 1.32) | 1.14 (0.93 to 1.38) | 0.70 (0.62 to 0.79) | 0.83 (0.72 to 0.94) |
| Age group (years)[f] | | | | | |
| 40–49 | 0.83 (0.80 to 0.86) | 1.0 (0.95 to 1.04) | 1.15 (1.06 to 1.24) | 0.77 (0.73 to 0.81) | 0.83 (0.79 to 0.88) |
| 50–59 | 1.0 (reference) | 1.0 (reference) | 1.0 (reference) | 1.0 (reference) | 1.0 (reference) |
| 60–69 | 1.21 (1.17 to 1.26) | 0.98 (0.94 to 1.02) | 0.80 (0.74 to 0.86) | 1.28 (1.21 to 1.34) | 1.29 (1.22 to 1.36) |
| 70–79 | 1.21 (1.16 to 1.27) | 0.87 (0.83 to 0.92) | 0.66 (0.60 to 0.73) | 1.02 (0.96 to 1.08) | 1.44 (1.35 to 1.54) |
| ≥80 | 0.88 (0.81 to 0.95) | 0.64 (0.58 to 0.71) | 0.39 (0.30 to 0.49) | 0.45 (0.41 to 0.49) | 1.17 (1.04 to 1.32) |

[a]Each column in the table represents a separate model for race/ethnicity and age group with the column header indicating the dependent variable classification with the reference being none

[b]Among women with mammogram encounters from January 1, 2019 to December 31, 2019 (n = 79,257 for race/ethnicity analysis; n = 79,983 for age analysis)

[c]Among women with mammogram encounters from June 2, 2018 (when EPIC started) through December 31, 2018 (n = 38,654 for race/ethnicity analysis; n = 39,035 for age analysis)

[d]Adusted for age category at 2019 encounter (race/ethnicity analysis) and median household income quartile (both analyses)

[e]Adjusted for age category at 2018 encounter (race/ethnicity analysis) and median household income quartile (both analyses)

[f]Age category in 2019 for analyses among women with screening encounters in 2019 (i.e. columns 2–5) and age category in 2018 for analyses among women with screening encounters in 2018 (i.e. column 6).

encounter in 2021 vs. none in 2020 or 2021 were similar in those 40–49 and 60–69 ($OR_{40-49}$ = 1.0, 95% CI 0.95 to 1.04; $OR_{60-69}$ = 0.98, 95% CI 0.94 to 1.02), and lower in those ≥70 ($OR_{70-79}$ = 0.87, 95% CI 0.83 to 0.92; $OR_{\geq 80}$ = 0.64, 95% CI 0.58 to 0.71) vs. those 50–59 years old. The odds of a triennial screening mammogram return encounter in 2022 vs. none in 2020–2022 was significantly lower in all three older age groups with ORs ranging from 0.39 to 0.80 and higher in those 40–49 years old (OR = 1.15, 95% CI 1.06 to 1.24) vs. those 50-59-year-olds. Finally, the odds of any screening encounter return during the pandemic years were significantly lower in 40–49 and ≥80 ($OR_{40-49}$ = 0.77, 95% CI 0.73 to 0.81; $OR_{\geq 80}$ = 0.45, 95% CI 0.41 to 0.49) and higher in those 60–69 years old (OR = 1.28, 95% CI 1.21 to 1.34) vs. those 50-59-year-olds (Table 3).

To evaluate the possibility that the observed differences in screening mammogram return encounters were pre-existing, we conducted analyses to determine whether there were race/ethnicity and age group differences in screening mammogram return encounters in 2019 among those screened in 2018 (from June 2, 2018 when EPIC started). The patterns were generally similar to or in the same direction as the 2020 results across race/ethnicity groups and age groups except for those ≥80 years old (Table 3).

## Discussion

Our findings suggest that the COVID-19 pandemic affected breast cancer screening mammogram encounters for an approximately two-month period with no statistically significant variation in the proportion of missed or delayed visits by race/ethnicity or age group. However, our

model results suggest that screening avoidance may have predated the official stoppage instituted by the healthcare system on 3-23-2020 in some groups. When examining evidence for pandemic-associated disparities in return screening mammogram encounters, we found that among women screened in 2019 the odds of a return screening encounter was similar in 2020 and lower in 2021 and in all three years of the pandemic (2020 to 2022) in NH Black vs. NH White women. Hispanic and NH Asian vs. NH White women had lower odds of a return screening encounter during 2020 and in all three years of the pandemic. With respect to age group, women ages 60–79 years old had higher odds of a return screening encounter during the first year of the pandemic with those 60–69 years old also having higher odds of any screening encounter in all three years of the pandemic, while those 40–49 and ≥80 years old had a lower odds vs. women 50–59 years old for both analyses. However, we note that our sensitivity analysis suggests that pre-existing disparities by race/ethnicity cannot be excluded at least in the first year of the pandemic, which indicates that these patterns may not be fully due to the pandemic.

Our findings modeling weekly screening mammogram counts are generally consistent with those reported in a systematic review including 74 studies on the pandemic's effect on mammograms [10]. Among 17 U.S. studies conducted within healthcare systems or single departments/institutions; most had follow-up ending in 2020 with one study ending on 4-30-2021. The percent decline in the number of mammographic screens in the pandemic period compared to a pre-pandemic period ranged from 9.8% to 99.8%; however, the length of the period examined varied markedly. Our results are generally consistent in magnitude with studies that examined drops in screening during a similar time frame with reported drops ranging from 77.7% (3-2-2019 to 6-2-2019 vs. same period in 2020) to 99.8% (3-19-2019 to 5-9-2019 vs. same period in 2020) [10, 12, 14, 22–28].

Several U.S. studies have reported screening mammograms drops in racial/ethnic minority groups that are slightly higher than in Whites [10, 12, 14, 26, 29–33]. For example, Degroff et al. [12] reported a 90% drop in screening in Blacks in April 2020 vs. an 87% drop in Whites, and Velazquez [30] reported a 14% drop in Black or African-American women in the first shutdown period of 2-1-2020 to 5-31-2020 vs. a 9% drop in Whites. In our analysis, we did not observe statistical evidence for larger drops in racial/ethnic minority groups than in Whites during the predicted pandemic-affected period.

One study reported greater drops in the number of screens in those <50 years old than in those 50–64 years old and increases in screens in those ≥65 years old when comparing 2020 to 2019 in women in Washington state [13]. Another U.S. study showed the lowest drops in screening numbers among women seen at a San Francisco, CA safety net hospital who were ≥70 years old compared to those who were younger in the period of 2-1-2020 to 5-31-2020 (the first shutdown period). However, this pattern was reversed during a second shutdown period from 12-1-2020 to 1-31-2021 vs. the pre-pandemic period with the highest drops in screening in those ≥70 years [30].

Our modeling results should be considered within the context of the shutdown of mammogram screening at the facilities included in this report that began March 23, 2020 [17]. Although we could not locate an official date when screening mammograms resumed, newspaper reports [34] suggest resumption in early May, which is in accordance with our data. While the predicted end of the pandemic-affected period demonstrates a lag of around two weeks, mammogram screening appears to have recovered quickly.

In our attrition analyses that examined the odds of returning for a mammogram during the pandemic years (2020, 2021, and 2022) among those with screening encounters in 2019, we observed a lower odds of returning for a screening mammogram in Hispanic and NH Asian women compared to NH White women in the first year of the pandemic and in any year of the

pandemic. NH Black or African American women had a lower odds of a return biennial screening encounter in 2021 or in any of the pandemic years 2020–2022 than NH White women. However, they were significantly more likely to have a triennial screening encounter than non-Hispanic White women. Although these data could indicate possible longer-term screening disparities associated with the pandemic, our sensitivity analysis of a return screening encounter in 2019 among those screened in 2018 suggests a possible pre-existing pattern. Evidence for a potential pandemic-associated exacerbation of disparities was observed in NH Asian women where the odds of a return screening encounter in 2020 among those screened in 2019 was lower than the odds of a return screening encounter in 2019 among those screened in 2018. This result is in line with a study reporting a greater pandemic-associated effect on drops in the prevalence of screening mammograms among NH Asian women [35]. A prior study in 2018 in a study population overlapping with ours also provides evidence supporting pre-existing disparities [36]. Additional research is needed to quantify the contribution of the pandemic to racial/ethnic disparities in annual screening and whether stage at diagnosis varied by screening status during the pandemic.

Evidence for a lower rate of return for annual screening mammograms among racial/ethnic minority groups was suggested in a study by Miller et al. including pre-pandemic data from 2019 and pandemic data from 2020. Miller et al. reported an OR of 0.88 (95% CI 0.80 to 0.98); however, when adjusting for socioeconomic variables, the result was of similar magnitude at 0.9 but no longer significant. With regard to age, the Miller et al. study showed that among women screened in 2019, the odds of a return screen in 2020 was higher at 1.29 (95% CI 1.19 to 1.41) in women ≥65 vs. <65 years old [32]. Similarly, we observed that older women from 60 to 79 years old in 2019 had a higher odds of return encounters in 2020 than women ages 40 to 59 years old. Women ≥80 years old in 2019 had the lowest odds of a return screening encounter during the pandemic years. However, we could not account for factors such as screening guidelines that indicate screening can be discontinued if life expectancy is <10 years [7] or vital status in our analyses since this is not systematically collected, which could partially explain this result.

Our study has strengths and limitations. All prior studies to our knowledge heuristically selected time period ranges for pre- and post-pandemic comparisons. Unlike other studies using count data comparing post- to pre-pandemic counts, a notable strength of the current study is that we used a model to predict the time period within a range that we specified where the pandemic was potentially in effect overall and by sociodemographic subgroups. Our modeling approach incorporates information from time periods outside of the screening mammogram pause to better understand the variability in week-to-week screening numbers and exposure and uses that to predict what would have happened absent a pause in screenings. Therefore, our modeling approach is less susceptible to bias caused by non-random fluctuation of the number of encounters over time due to factors unrelated to the pandemic because it does not directly use the year before the pandemic (e.g. 2019 data) as a pre-pandemic period for comparison. However, modeling changes in counts over time has some limitations. We could not account for the denominator of women eligible for screening each week that would allow for comparison of how the incidence of screening changed during the pandemic using a comparison period before the pandemic among the population of women "at risk" for screening.

A limitation of our attrition analysis is that we could not account for vital status or mammograms received outside of the healthcare system from which we obtained data as reasons for not returning for an annual screening mammogram because these variables are not systematically collected. If a higher percentage of racial/ethnic minority women than NH White women died or received their mammograms outside of the healthcare system included in this

study during the pandemic, our ORs <1 may be biased away from the null resulting in overestimates of disparities in screening between NH White and racial/ethnic minority women. In addition, we also note that although mammograms are recommended annually by the healthcare system from which our data were obtained [37], national guidelines vary by age, risk factors, and the guideline recommending body [7, 9]. In addition, patients and providers within the same healthcare system may have adopted practices differing from those recommended by the healthcare system on public-facing websites [37]. We could not account for this difference in our models as well as other potential confounding factors such as comorbidities that may affect return screening encounters. Any differences in the odds of screening must therefore be considered within the context of these factors. Finally, we note that this study is based on data collected by one healthcare system and is not population-based.

In conclusion, our results suggest a short-term effect of the pandemic on mammogram screening of about two months with no evidence of disparities. The majority of this window overlaps the cancellation of elective procedures. Racial and ethnic disparities were observed in returns for annual screening mammogram encounters; however, our sensitivity analysis suggested that this pattern may be at least partially pre-existing. These data suggest extra outreach to these groups to increase adherence to recommended screening frequency guidelines is needed whether or not a pandemic is occurring. These results may inform future pandemic planning and the need for continued efforts to eliminate disparities in annual screening rates.

## Supporting information

**S1 File. These are the supplementary methods.**
(DOCX)

**S2 File. R code file for attrition analysis including all data management and logistic regression models.**
(RTF)

**S3 File. Code file for data management for the Bayesian state space model results.** It includes wrangling data from an administrative data format detailing individual instances of claims representing routine screening mammograms to weekly counts by group as well as Bayesian model specification and modeling using Stan.
(RTF)

**S4 File. Code file that generates an HTML document of results presented in the paper.**
(RTF)

**S5 File. Code file for the model fit objects.** It takes them and wrangles them into the appropriate output for tables and figures.
(RTF)

**S1 Data. These are the zip codes in the catchment area (Zipcodes sheet) and the zip codes from Social Explorer that overlap with the zip codes in the catchment area (ACSdata sheet).**
(XLSX)

## Author Contributions

**Conceptualization:** Kimberly J. Johnson, R. J. Waken, Justin M. Barnes.

**Data curation:** Kimberly J. Johnson.

**Formal analysis:** Kimberly J. Johnson, Caitlin P. O'Connell, R. J. Waken.

**Funding acquisition:** Kimberly J. Johnson.

**Investigation:** Kimberly J. Johnson.

**Methodology:** Kimberly J. Johnson, R. J. Waken.

**Supervision:** Kimberly J. Johnson.

**Visualization:** Kimberly J. Johnson, R. J. Waken.

**Writing – original draft:** Kimberly J. Johnson, R. J. Waken.

**Writing – review & editing:** Kimberly J. Johnson, Caitlin P. O'Connell, R. J. Waken, Justin M. Barnes.

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
