## [Decision Letter · Decision Letter 0]

30 Jan 2024

PONE-D-23-31782Impact of COVID-19 pandemic on breast cancer screening and diagnoses in a large Midwestern United States academic medical centerPLOS ONE

Dear Dr. Johnson,

Thank you for submitting your manuscript to PLOS ONE. After careful consideration, we feel that it has merit but does not fully meet PLOS ONE’s publication criteria as it currently stands. Therefore, we invite you to submit a revised version of the manuscript that addresses the points raised during the review process.

 Please submit a point-by-point response that addresses each of the reviewers' excellent comments. In particular, note the questions about characteristics of the underlying population, differences between screening and diagnostic exams, and recommendations for annual vs biennial screening. Understanding each of these components affects how the reader interprets your results.

We look forward to receiving your revised manuscript.

Kind regards,

Erin J A Bowles

Academic Editor

PLOS ONE

Journal Requirements:

2. Thank you for stating the following financial disclosure: "This work was supported by a just-in-time award through NIH CTSA Grant #UL1 TR002345".

3. Please expand the acronym "NIH" as indicated in your financial disclosure) so that it states the name of your funders in full.

Reviewers' comments:

Reviewer's Responses to Questions

**Comments to the Author**

1. Is the manuscript technically sound, and do the data support the conclusions?

Reviewer #1: Yes

Reviewer #2: Yes

2. Has the statistical analysis been performed appropriately and rigorously? 

Reviewer #1: Yes

Reviewer #2: Yes

3. Have the authors made all data underlying the findings in their manuscript fully available?

Reviewer #1: No

Reviewer #2: No

4. Is the manuscript presented in an intelligible fashion and written in standard English?

Reviewer #1: Yes

Reviewer #2: Yes

5. Review Comments to the Author

Reviewer #1: Thank you for interesting paper. The paper gives a nice overview of the impact of the COVID-19 pandemic on breast cancer screening and diagnoses. The authors show a decrease in the number of screening mammograms between 3-15-2020 and 5-24-2020, returning to pre-pandemic levels thereafter. Additionally, they show that Hispanic and non-Hispanic Asian women have a higher odds of not being screened in 2020/2020, 2021 or 2022/2020, 2021 and 2022 compared to non-Hispanic White women. Identifying which groups are less likely to get a mammogram is a first important step to increase the number of women who go for a screening mammogram. The analyses are well performed and the results are nicely presented. I still have a few points I would like to address.

Introduction. Please explain more about the breast cancer screening program of the United States, as this screening program is completely different from the ones in Europe.

Introduction. You write “Before recent changes in screening recommendations [7] and during the pandemic, annual or biennial mammogram screening for breast cancer in the United States was recommended for women ages 45 or 50 and older who are at average risk”. When is a women at “average risk”? And there is a huge difference between annual and biennial mammogram screening. When do they recommend women to undergo annual screening and when biennial? Related to this: I read in the discussion that annual screening is recommended by the healthcare system from which your data is obtained. This is very important to mention in the introduction, so the readres understand why you looked at annual, and not biennial, screening.

Methods: Could you please explain a bit more about those women who ‘were included on a registry that tracks wellness markers (e.g. mammmograms, annual visits)’. What kind of women are these? Is everyone included in this registry or are those a particular type of women (do the included women have the same age, socioeconomic status, breast cancer risk, etc.?).

Methods: For me the most difficult part to understand is how you have decided that the women visit the screening program annually. You want to investigate whether women return for an annual screening. But should you not know how many women do have an annual mammogram in the first place? Maybe women do not return for an annual screening because they chose for biennial screening.

Method: Why did you choose to only use results from 2019 to predict counts of screening mammogram encounters, and not for instance from 2017-2019?

Results: Only a minor detail, but in title of Table 1 you write 319,492 (with a comma after the first three numbers) and in the main body of the table you do not use a comma to separate the first two numbers from the last three numbers.

Results: Was marital status also collected in EPIC?

Results: You write “The only differences in the predicted pandemic-affected period by age group were for those ≥70 years starting on 3-15-2020” This is exactly the same date as the date for the total population. Should you than not write that those <70 have a difference in the predicted pandemic-affected period?

Results: In Table 2 you mention the “pandemic period last week” for each age group. How can this be 5-17-2020 for each age group, while this is 5-24-2020 for everyone combined? This seems odd.

Discussion: What are the odds for developing breast cancer in Hispanic and non-Hispanic Asian women compared to non-Hispanic White women. The first two mentioned groups have a lower odds of going back for an annual screen. However, this does not has to be a problem when their odds of getting breast cancer is also lower. Would you say the lower odds of getting an annual screen is worrisome? Should action be undertaken?

Reviewer #2: Overall Summary:

This single institution study estimates the proportion of missed/delayed screening mammograms that did not occur during the pandemic and the subsequent racial differences in return to screening mammography in the subsequent years. Racial and ethnic disparities in subsequent return to screening were identified, however, sensitivity analysis suggests these were pre-existing disparities.

Strengths:

This retrospective study incorporates a new analytic approach to estimate the missed/delays screening during the pandemic. Topic is very relevant as identifying disparities in return to screen may help create targeted interventions.

Weaknesses:

1. The clinical relevance of predicting the pandemic-affected period and differences by race is not well communicated.

2. Additional details in the methods are necessary to better understand whether capture of return screening mammograms outside of the institution was attempted/available. The possibility of patients seeking screening mammograms outside of the institution may cause disparities that are not actually present or that exist in smaller degrees, but this is not addressed in the discussion.

3. Many additional variables likely influence return to screen, such as provider, clinic, comorbidity, and these are not included in the study as confounders.

Specific Comments:

Title/Abstract

1. Title mentions diagnoses but no part of the manuscript addresses diagnosis of breast cancer

Introduction

2. Perhaps highlight the need for this new analytic approach

Methods and Materials

3. Study population: Who are the women included in the registry tracking wellness markers? Are they different from other patients at BJC? This and exclusions due to residing in zip codes outside of catchment, and several racial and ethnic minority groups have the potential to introduce a lot of selection bias that is not addressed.

4. How were dates selected for the pandemic indicator variable? Why did this not align with when the institution stopped screening mammograms on 3/23/2020?

5. How was it handled if the patient returned for diagnostic imaging rather than screening mammography? While a “screening mammogram” they did return for imaging.

6. In describing the variable to categorize screening patterns among women screened in 2018, it would be easier to read if “returned” and “did not return” were in quotation marks

7. How was missing median household income handled and marital status handled?

8. It should be clearly defined how it was determined whether a woman returned for a screening mammogram in the attrition analysis. Was this based on completed encounters in EPIC? Was encounter information from outside institutions available/included? If patient reported a screen elsewhere, was this included?

9. Would state why the 2018 comparison period started in June in the methods

Results

10. Subheading would help the reader

11. “The only differences in the predicted pandemic-affected period by age group were for those ≥70 years starting on 3-15-2020” << I don’t understand this sentence. The pandemic period start was also 3/15/2020

12. Overall, the sentences describing the primary attrition results should be restructured because they are currently difficult to read.

13. In describing results, it would be helpful to explicitly state the reference point for missed or delayed appointments (i.e. compared to predicted) at least once

Discussion

14. The first paragraph should talk about the pandemic attrition results before the sensitivity analysis pre-existing attrition

15. Why do you think racial/age disparities were not observed during the pandemic affected period in your study?

16. “we observed higher odds of not returning in Hispanic and non-Hispanic Asian women compared to non-Hispanic white women in both 2020 and 2021 and in all three years for all three minority groups” < “in both 2020 and 2021” is not clear here because there were three outcomes but not clear if this part is referring to 1 or 2 of the outcomes

17. Women >80 years old are also probably not returning to screening because guidelines only recommend screening until age 74 or until life expectancy is <10 years. So, death isn’t the only reason they would stop screening

18. “However, it seems unlikely that this bias would result in non-differential misclassification by race/ethnicity” < Is this sentence what you mean to say? But wouldn’t lack of vital status lead to differential misclassification since average age at death varies by patient race?

Tables/Figures

19. Figure 1: These types of figures are easier to understand with there are separate exclusion boxes off to the side; this allows the final box actually describes what the cohort is instead of highlighting the last exclusion; and additional level of the figure would also be helpful so the time series cohort is distinguished from the Attrition Cohort

20. P-values should be added to Table 2, instead of just in the text

21. Time series figures should be updated to improve the clarity of the axis labels; “mean” is not adequate

References

22. Replace reference 8 with: Oeffinger KC, Fontham ET, Etzioni R, et al. Breast cancer screening for women at average risk: 2015 guideline update From the American Cancer Society. JAMA. 2015;314(15):1599-1614.

23. Replace reference 9 with: Ann Intern Med. 2016;164:279-296. doi:10.7326/M15-2886

24. Reference 11 and 32 are the same

25. Reference 13 and 19 are the same

26. Reference 23 and 29 are the same

6. PLOS authors have the option to publish the peer review history of their article (what does this mean?). If published, this will include your full peer review and any attached files.

Reviewer #1: **Yes: **Anouk H. Eijkelboom

Reviewer #2: No

---

## [Author Response · Author response to Decision Letter 0]

2 Apr 2024

Please see attached word document response and below with table missing

Point-by-point Response to Reviewers’ Comments

Reviewer #1: Thank you for interesting paper. The paper gives a nice overview of the impact of the COVID-19 pandemic on breast cancer screening and diagnoses. The authors show a decrease in the number of screening mammograms between 3-15-2020 and 5-24-2020, returning to pre-pandemic levels thereafter. Additionally, they show that Hispanic and non-Hispanic Asian women have a higher odds of not being screened in 2020/2020, 2021 or 2022/2020, 2021 and 2022 compared to non-Hispanic White women. Identifying which groups are less likely to get a mammogram is a first important step to increase the number of women who go for a screening mammogram. The analyses are well performed and the results are nicely presented. I still have a few points I would like to address.

1. Introduction. Please explain more about the breast cancer screening program of the United States, as this screening program is completely different from the ones in Europe.

Response: Thank you for this comment. We revised our description of screening recommendations in the United States to increase its clarity in paragraph 2 of the Introduction section.

2. Introduction. You write “Before recent changes in screening recommendations [7] and during the pandemic, annual or biennial mammogram screening for breast cancer in the United States was recommended for women ages 45 or 50 and older who are at average risk”. When is a women at “average risk”? And there is a huge difference between annual and biennial mammogram screening. When do they recommend women to undergo annual screening and when biennial? Related to this: I read in the discussion that annual screening is recommended by the healthcare system from which your data is obtained. This is very important to mention in the introduction, so the readers understand why you looked at annual, and not biennial, screening.

Response: We thank the reviewer for these comments that inspired us to reexamine our analytic approach with consideration for variable U.S. screening guidelines. There is no standardized approach to breast cancer risk assessment that qualifies someone as average risk in the United States [1]. To the best of our knowledge, “average risk” is generally a woman without factors putting her at high risk (e.g. a germline pathogenic mutation in the BRCA1 or 2 genes [2]). There are two major guideline-recommending bodies for breast cancer screening, the United States Preventive Services Task Force (USPSTF) and the American Cancer Society (ACS) with the frequency of screening varying by age and guideline-recommending body. A table of their recommendations by age for “average risk” women is below. To address the reviewer’s last comment about the healthcare system from which these data were obtained, we conducted additional analyses to account for biennial screening as guideline concordant screening, we did keep the note about the healthcare system’s public-facing recommendation in the discussion because the frequency of screening also involves individual patient choice and provider recommendation, which may vary from those of the healthcare system overall. We have updated the Introduction paragraph 2 to be clearer about guidelines and note the biennial analysis in the Methods variables sub-section.

[See table in PDF]

3. Methods: Could you please explain a bit more about those women who ‘were included on a registry that tracks wellness markers (e.g. mammograms, annual visits)’. What kind of women are these? Is everyone included in this registry or are those a particular type of women (do the included women have the same age, socioeconomic status, breast cancer risk, etc.?).

Response: The registry tracks annual visits and mammograms among individuals seen in the BJC healthcare system. The registry does not select based on age, socioeconomic status, breast cancer risk, or any other variable. This information is detailed in the Supplementary Methods Section I.

4. Methods: For me the most difficult part to understand is how you have decided that the women visit the screening program annually. You want to investigate whether women return for an annual screening. But should you not know how many women do have an annual mammogram in the first place? Maybe women do not return for an annual screening because they chose for biennial screening.

Response: Thank you for this comment. We appreciate that some women may choose biennial screening, and this is still guideline-concordant screening. Because national screening guidelines vary as detailed in the introduction, we have revised our attrition analysis to include an outcome category that accounts for women screened annually, biennially, triennially, and at any time during the pandemic:

1. The odds of a screening encounter return in 2020 vs. no screening encounter return in 2020 by race/ethnicity and age group among those screened in 2019. This analysis provides results for those who undergo annual screening vs. no screening in the first year of the pandemic when facilities shut down for a short period by race/ethnicity and age group.

2. The odds of a screening encounter return in 2021 vs. no screening encounter return in 2020 or 2021 by race/ethnicity and age group among those screened in 2019. This analysis accounts for those who undergo biennial screening vs. no screening in the first two years of the pandemic by race/ethnicity and age group. 

3. The odds of a screening encounter return in 2022 vs. no screening encounter return in 2020-2022 by race/ethnicity and age group among those screened in 2019. This analysis provides results for those who undergo triennial screening vs. no screening in the first three years of the pandemic by race/ethnicity and age group.

4. The odds of any screening encounter return in 2020-2022 vs. no screening encounter return in 2020-2022 by race/ethnicity and age group among those screened in 2019. This analysis provides results for those who underwent any screening during the pandemic vs. no screening by race/ethnicity and age group.

5. The odds of a screening encounter return in 2019 vs. no screening encounter return in 2019 by race/ethnicity and age group among those screened in 2018 (this is a sensitivity analysis). This analysis provides results to gauge whether observed disparities/patterns may have existed before the pandemic.

We have explained our rationale and variable definitions in the revised Variables sub-section of the Methods section. We changed to predicting the odds of a screening encounter vs. the odds of no screening encounter because we felt it increased the clarity of communication. This change is reflected in Table 3 and throughout the text of the revised manuscript.

5. Method: Why did you choose to only use results from 2019 to predict counts of screening mammogram encounters, and not for instance from 2017-2019?

Response: Thank you for this question. We do not have EPIC data from 2017 and only the last 6 months of 2018, when EPIC was implemented, which limited our ability to conduct more in-depth analyses. We have made this clear in the first paragraph of the Study Population sub-section of the Methods section and the Attrition Analysis sub-section of the Methods section.

6. Results: Only a minor detail, but in title of Table 1 you write 319,492 (with a comma after the first three numbers) and in the main body of the table you do not use a comma to separate the first two numbers from the last three numbers.

Response: Thank you for this comment. Commas have been added to the numbers in the table and the column header.

7. Results: Was marital status also collected in EPIC?

Response: Marital status information from EPIC data is provided in Table 1. 

8. Results: You write “The only differences in the predicted pandemic-affected period by age group were for those ≥70 years starting on 3-15-2020” This is exactly the same date as the date for the total population. Should you than not write that those <70 have a difference in the predicted pandemic-affected period?

Response: Thank you for pointing out that the comparison in this statement was a little unclear, we revised this statement as suggested by the reviewer in the second section of the Results titled Model predicted periods where the pandemic affected mammogram encounters. We explain why this may occur in this section.

9. Results: In Table 2 you mention the “pandemic period last week” for each age group. How can this be 5-17-2020 for each age group, while this is 5-24-2020 for everyone combined? This seems odd.

Response: Thank you for pointing out that the way the model estimated the acute drop in screenings during the pandemic was not made clear – we have added an additional explanation in both the results under Model predicted periods where the pandemic affected mammogram encounters and in the first paragraph of the Methods Bayesian State Space Modeling analysis sub-section.

10. Discussion: What are the odds for developing breast cancer in Hispanic and non-Hispanic Asian women compared to non-Hispanic White women. The first two mentioned groups have a lower odd of going back for an annual screen. However, this does not has to be a problem when their odds of getting breast cancer is also lower. Would you say the lower odds of getting an annual screen is worrisome? Should action be undertaken?

Response: Thank you for this point. The screening guidelines from the ACS and the USPSTF do not stratify their recommendations by race so although certain groups may have lower risk, annual or biennial screening is still recommended. This is an interesting point but a discussion of whether our finding could translate to increased risk for later-stage breast cancer. We did not investigate this in this paper, but we have added a point to the last sentence of discussion paragraph 6 to emphasize the need for these data.

Reviewer #2: Overall Summary:

This single institution study estimates the proportion of missed/delayed screening mammograms that did not occur during the pandemic and the subsequent racial differences in return to screening mammography in the subsequent years. Racial and ethnic disparities in subsequent return to screening were identified, however, sensitivity analysis suggests these were pre-existing disparities.

Strengths:

This retrospective study incorporates a new analytic approach to estimate the missed/delays screening during the pandemic. Topic is very relevant as identifying disparities in return to screen may help create targeted interventions.

Weaknesses:

1. The clinical relevance of predicting the pandemic-affected period and differences by race is not well communicated.

Response: Thank you for this suggestion. We have added details about how the period in which acute drops could occur by group to the methods section to the last section in the Bayesian State Space methods first paragraph. In this case, if we specify a model that does not allow there to be a difference in groups, the model structure allows no difference between the groups by design.

2. Additional details in the methods are necessary to better understand whether capture of return screening mammograms outside of the institution was attempted/available. The possibility of patients seeking screening mammograms outside of the institution may cause disparities that are not actually present or that exist in smaller degrees, but this is not addressed in the discussion. 

Response: Thank you for this important point regarding missing data (i.e. screening mammograms performed outside of the BJC healthcare system). These are not systematically captured in EPIC and depend on patients sharing their data across institutions. We have acknowledged this limitation in the limitations section of the Discussion section and changed our terminology and table 3 and in the text (screening encounter return) to more accurately reflect what we could measure.

3. Many additional variables likely influence return to screen, such as provider, clinic, comorbidity, and these are not included in the study as confounders.

Response: The reviewer is correct; however, these were not included in our data. Not adjusting for these variables would primarily affect the screening attrition analysis. Since these women were all seen in one healthcare system that recommends annual mammography screening and we conducted analyses to account for patient choice and provider recommendations to screen annually or biennially, we do not think that adjusting for provider or clinic would change overall conclusions. Concerning comorbidities, guidelines only address ceasing screening if life expectancy is less than 10 years with no specific recommendations for screening among people with comorbidities. We have added this limitation to the limitations section of the Discussion section. We also note that women who have a medical exclusion listed as a reason for not getting a mammogram were excluded. This is detailed in the Supplementary Methods Section I.

Specific Comments:

Title/Abstract

1. Title mentions diagnoses but no part of the manuscript addresses diagnosis of breast cancer

Response: Thank you for this comment. We removed diagnosis from the title as that analysis is not part of this manuscript.

Introduction

2. Perhaps highlight the need for this new analytic approach 

Response: Thank you for the suggestion. We have added justification for using a forecasting/backcasting predictive approach to quantifying the number of missed screenings by group, which is typically referred to as a Brownian bridge. This is a multivariate extension. This justification is now included in the “Pandemic effects on encounters sub-section” of the Methods section.

Methods and Materials

3. Study population: Who are the women included in the registry tracking wellness markers? Are they different from other patients at BJC? This and exclusions due to residing in zip codes outside of catchment, and several racial and ethnic minority groups have the potential to introduce a lot of selection bias that is not addressed.

Response: The registry tracking wellness markers includes all women seen in the BJC system. They are not different from other patients at BJC except from those patients who were specifically excluded as noted in the Supplementary methods. We have added a sentence to the limitations section to note that the study is not population-based. We have also added the limitation of not being able to capture screening mammograms outside of the BJC system. 

4. How were dates selected for the pandemic indicator variable? Why did this not align with when the institution stopped screening mammograms on 3/23/2020?

Response: Thank you for the opportunity to clarify how this model approaches predicting the drop in screening visits. Additional justification has been added to the Methods Pandemic effects on encounters section. The model doesn’t dictate a single time point at which screenings stopped but rather lets the data choose as healthcare avoidance in addition to institutional decisions that also contributed to the drop in screenings. To reduce the computational burden, we allow the model to choose any day between 3/1/2020 and 4/15/2020 for the dropoff to start.

5. How was it handled if the patient returned for diagnostic imaging rather than screening mammography? While a “screening mammogram” they did return for imaging.

Response: These are marked separately in the database. We did not include diagnostic mammograms in our analyses. This is noted in the revised Figure 1 flowchart exclusions and in Section II of the Supplementary Methods.

6. In describing the variable to categorize screening patterns among women screened in 2018, it would be easier to read if “returned” and “did not return” were in quotation marks

Response: We changed this section and analysis to increase the clarity of the results as described in our response to Reviewer 1’s comment 4.

7. How was missing median household income handled and marital status handled?

Response: The marital status variable was only used in Table 1. The missing data on this variable is small with <1% of encounters having unknown or

---

## [Editor Report · Decision Letter 1]

23 Apr 2024

Impact of COVID-19 pandemic on breast cancer screening in a large Midwestern United States academic medical center

PONE-D-23-31782R1

Dear Dr. Johnson,

We’re pleased to inform you that your manuscript has been judged scientifically suitable for publication and will be formally accepted for publication once it meets all outstanding technical requirements.

Kind regards,

Erin J A Bowles

Academic Editor

PLOS ONE
---

## [Editor Report · Acceptance letter]

30 Apr 2024

PONE-D-23-31782R1 

PLOS ONE

Dear Dr. Johnson, 

I'm pleased to inform you that your manuscript has been deemed suitable for publication in PLOS ONE. Congratulations! Your manuscript is now being handed over to our production team.

Kind regards, 

on behalf of

Dr. Erin J A Bowles 

Academic Editor

PLOS ONE